# Hiding Information in Digital Images Using Ant Algorithms

**DOI:** 10.3390/e25070963

**Published:** 2023-06-21

**Authors:** Mariusz Boryczka, Grzegorz Kazana

**Affiliations:** 1Institute of Computer Science, University of Silesia in Katowice, Bedzińska 39, 41-200 Sosnowiec, Poland; 2SkyGate, Chris Parjaszewski, Rynek 6, 44-100 Gliwice, Poland; kazana.grzegorz@gmail.com

**Keywords:** steganography, digital image, ant algorithms, high capacity, low distortion

## Abstract

Stenographic methods are closely related to the security and confidentiality of communications, which have always been essential domains of human life. Steganography itself is a science dedicated to the process of hiding information in public communication channels. Its main idea is to use digital files or even communication protocols as a medium inside which data are hidden. The present research aims to investigate the applicability of ant algorithms in steganography and evaluate the effectiveness of this approach. Ant systems could be employed both in spatial and frequency-based image steganography. The combination of frequency domain and optimization method to increase robustness is used, and an integer wavelet transform is performed on the host image. ACO optimization is used to find the optimal coefficients describing where to hide the data. The other method utilizes ACO to determine the optimal pixel locations for embedding secret data in the cover image. ACO is also used to detect complex regions of the cover image. Afterward, the least-significant-bits (LSB) substitution is used to hide secret information in the detected complex regions’ pixels. Our study focuses on optimizing two mutually exclusive features of steganograms—high capacity and low distortion. An attempt was made to use ant systems to select areas of digital images that allow the greatest amount of information to be hidden with the least loss of image quality. The effect of variants of the ant system and its parameters on the quality of the results obtained was also investigated, and the final effectiveness of the proposed method was evaluated. The results of the experiments were compared with those published in related articles. The proposed procedures proved to be effective and allowed the embedding of large amounts of data with relatively little impact on image quality.

## 1. Introduction

Security and confidentiality of communications have always been essential in human life. Historically, the need for the exchange of classified information has primarily concerned rulers and politicians. Over the previous decades, this trend began to change. Due to the ubiquitous digitization of human life and modern forms of communication, the need for means of exchanging classified information affects everyone.

Data hiding is one method of protecting information from unauthorized access. It allows data to be hidden in a different way than under normal circumstances, which can make it more difficult for third parties to attempt to breach it. Data hiding is especially important for sensitive information, such as medical or financial data, not to mention business-critical or national security data, which can be valuable targets for cybercriminals. To ensure the best protection of data, various methods of protection should be used, including those that allow them to be hidden.

Apart from encrypting the contents of the message, attempts at hiding the occurrence of communication itself seem to be reasonable and potentially advantageous in ensuring communication privacy. This is the domain of steganography. The vast volume of information that is transmitted over communication networks gives hope and opens the door for its use in information hiding.

In steganography, the concept of security (confidentiality) is very often used as one of the main criteria for its quality [1]. The confidentiality criterion refers to the ability of the system to ensure that the hidden information remains secret and cannot be easily detected or accessed by unauthorized individuals. This criterion primarily focuses on protecting the confidentiality of the hidden data. On the other hand, the image distortion criterion in steganography pertains to the extent to which the process of embedding secret information into an image alters or distorts the original image. The goal is to minimize the visible or perceptible changes introduced by the steganographic algorithm. Image distortion is an important factor because, if the changes are too noticeable, it can raise suspicion and increase the chances of detection by unintended recipients or attackers. Both the confidentiality criterion and image distortion criterion are significant considerations in the design and evaluation of steganographic systems.

In general, combining steganography with cryptography can provide an additional level of security and confidentiality for transmitted information. There are several ways in which the two can be combined [2,3,4]:Encryption before hiding: Cryptographic algorithms can be used to encrypt the message itself before it is hidden in the steganographic medium. This way, even if someone discovers the hidden message, they will not be able to read it without the right key to decrypt it.Hiding the cryptographic key: Steganography can be used to hide the cryptographic key in the media. The key can be placed in such a way that only a person who owns the medium and knows the proper hiding method will be able to retrieve it and use it to read the encrypted message.Combine steganography and cryptography in one process: One can also use both steganography and cryptography techniques in one process. For example, a message can be encrypted using a cryptographic algorithm, and then the encrypted message can be hidden in a steganographic medium. In this way, the message is protected by both cryptography and steganography.

It is important to remember that the combination of steganography and cryptography does not provide absolute certainty and security. The use of appropriate cryptographic algorithms and keys, as well as careful concealment of the message in the steganographic medium, is crucial. In addition, when combining the two fields, performance and the possibility of detection by third parties must also be considered.

The article aimed to study the applicability of ant algorithms in steganography and evaluate their effectiveness. The research process focused on the use of ant colony systems in the selection of regions of digital images, which allows for hiding the largest amount of information with the least quality degradation. The work analyzed and compared methods for the graphical representation of images. For the ant algorithms that were used to solve the problem posed, an interpretation of the pheromone trace, which is the main medium of ant communication, was given. Then an analysis of the obtained results was carried out in terms of the loss of quality of images with the assumed amount of hidden data. The final step was to refer the results of the research to thematically related works.

The remainder of the article is organized as follows. A brief description of the issues related to steganography, with particular emphasis on digital steganography, is considered in Section 2. It also includes digital image steganography. Section 3 discusses the ant algorithms which have been used to hide messages in digital images. In turn, Section 4 describes the proposed solution in detail and also presents graph image representation and pheromone trail interpretation. This is followed by a description of performed experiments (Section 5). Their goal was to verify fundamental assumptions, such as the validity of selecting complex image regions, testing the efficacy of different methods of constructing graphs and pheromone trail interpretation, quantitative assessment of steganogram quality degradation, and subjective perceivability of the introduced changes. The results of the experiments are discussed in Section 6. Finally, the conclusions are given in the last section.

## 2. Steganography

Steganography is a science dedicated to the process of concealing information in public communication channels. In order to highlight concerns and characteristics of steganographic techniques, the notion of cryptography is regularly quoted [5]. The interest of cryptography is enabling the transfer of information via public communication channels while preventing third parties from understanding the contents of the message. Currently, this is achieved by using shared keys known only to participants (symmetric cryptography) or by the use of public and private keys (asymmetric cryptography) [6,7]. In contrast to cryptography, the goal of steganographic techniques is to enable private communication without disclosing the fact of the communication itself.

### 2.1. Digital Steganography

As a result of the increase in multimedia applications of modern computers and the spread of broadband Internet, digital steganography is becoming more prevalent and efficient. Its main idea is the use of digital files or even communication protocols as the medium within which the data are hidden. An example of protocol-based steganography is concealing information in the control bits of TCP/IP frames, introducing precise delays between subsequent packets, or even introducing intentional packet losses [8].

A significantly simpler, yet much more mature, technique is the use of multimedia files, such as digital photos, audio, or video files. The reason for their attractiveness as a data-hiding medium originates from their abundance, large file size, and redundancy [9]. The latter feature is of great importance, since it implies that modifying a subset of the information included in the file does not produce evident changes. For example, slight adjustments of the bitwise information of specific pixel channels will not cause obvious changes even for an attentive observer. Similar observation can be made also for audio files—manipulation of component frequencies which fall out of human perception, below 20 Hz or above 20 kHz, will not be perceivable by a subjective recipient [10]. Another example of data hiding in music is so-called backmasking, in which the data are revealed only if the file is played in reverse. One of the first music bands which popularized such experiments were The Beatles.

### 2.2. Digital Image Steganography

Despite the fact that every kind of binary file can be used as a steganographic medium, the most attention was paid to digital images and photos. Even though the simplest techniques are the most popular, there has been a rise in subtle and sophisticated methods, which differ both in complexity and produced results.

#### 2.2.1. Spatial Techniques

When employing spatial transformation methods, the input image is treated as a set of points (pixels) placed in a two-dimensional coordinate system. One of the benefits of such methods is their intuitiveness and approachability. Yet they are not free from downsides, since such methods are highly sensitive to attacks that either compromise information privacy or distort the secret message [5].

The most prevalent spatial technique is the Least Significant Bit (LSB). Its application relies on the manipulation of bits that are in the least significant positions of image pixels. Depending on the acceptable quality loss, one may modify *n* least significant bits of each RGB channel. A particular case of LSB is 4LSB. Such a method implies that exactly 4 bits of each image byte will be used, which results in 50% volume utilization of the medium. The cost of such a significant volume is apparent quality loss and simplified steganalysis.

In order to lessen the detectability of image manipulation with simultaneous high steganography volume, numerous techniques have been proposed (e.g., Variable Significant Bit (VLSB) [11]).

In order to improve the perceived image quality and decrease the risk of communication detection, DaChun Wu and W. Tsai proposed the Pixel Value Differencing (PVD) technique [12]. One of its main assumptions is making the number of used bits dependent on the difference in intensity between subsequent pixels. When implementing PVD, pixels are read sequentially in a pairwise fashion in such a manner that pixels in each pair are each other’s neighbors. For each pixel pair, the luminance difference is calculated, and based on that the number of used pixels is obtained. The number of used bits in each pixel is proportional to the difference between them. As a result, the steganogram is characterized by a lower risk of detection accompanied by significant volume [12].

#### 2.2.2. Frequency Domain Techniques

An alternative approach to digital image steganography are techniques that rely on image representations in the frequency domain. Such methods consist of transforming the bitmap to a matrix of coefficients that depict the amplitude or intensity of a particular frequency band present in the image [13]. Low-frequency bands correspond to the perception of vague shapes, colors of objects, and the composition thereof. High-frequency bands define and allow for the detection of sharp edges, details, and complex textures [14]. Regardless of hiding confidential data or digital signatures guarding digital copyrights, the main goal of frequency domain techniques is preservation of high data-hiding medium quality, and resilience against manipulation and distortion by compression [15].

One of the transforms used in steganography applications is Discrete Wavelet Transform (DWT) and the corresponding transform which allows for lossless inverse transform: Integer Wavelet Transform (IWT) [16]. One of the features of IWT is partitioning images into sections that consist of high and low-frequency bands. The article “Lossless Data Hiding Using Integer Wavelet Transform and Threshold Embedding Technique” [16] describes a method that relies on the substitution of least significant bits of high-frequency band coefficients. Experimental results, when compared to other techniques, proved a significant increase in concealed data volume while maintaining the same peak-to-signal noise ratio.

Based on the quoted literature and described methods, tendencies related to image region selection may be noticed. Featured articles which centered around frequency domain image representation made the decision of hiding data either in regions characterized by low- [17] or high-frequency coefficients [18].

One can find further works concerning steganography. For example, Xu, Zhao, and Huang [19] proposed a steganography method based on texture synthesis and compression for video sequences, Jang at al. [20] proposed a steganography method based on generative adversarial networks (GANs) for digital images, Liu at al. [21] proposed an image steganography scheme based on generalized Gaussian distribution and orthogonal matching pursuit (OMP), and Gupta and Rawat [22] proposed a novel image steganography approach based on multi-layer perceptron (MLP) and convolutional neural network (CNN). However, the work volume does not allow for a detailed discussion of these items.

## 3. Ant Systems

Ant systems (ASs) are metaheuristics that can be used in solving difficult optimization tasks. Their idea originates from a 1991 article written by M. Dorgio, V. Maniezzo, and A. Colorni [23]. Its authors proposed a concept and possible application of an algorithm that was modeled after behavior patterns of ants foraging for food. Since then, multiple variations and improvements to the algorithm were proposed. The idea was also applied to a large variety of problems from different domains.

The main principles that govern the behavior of an ant system were inspired by observations of ants’ behavior while exploring the environment near the anthill in order to find food. Initially, each ant wanders the surroundings of the anthill in a chaotic and random manner. In the case of finding the food, the ant will head back to the anthill. On its way back, the ant carrying food will lay a pheromone trail, which will allow for navigating back to the source of food. During a random exploration of the environment, both the ant which found the food and the rest of the colony members will favor routes that are marked by the pheromone trail. If the food source has not yet been exhausted, each consecutive ant will strengthen the trail. Such behavior produces a positive feedback loop effect since routes that were chosen by the ants will become increasingly preferable.

A crucial aspect of laying the pheromone trail is its evaporation over time. Such property is essential to a convergence of routes taken by ants to the shortest possible path. Authors of the algorithm summarized its fundamental characteristics in the following list [23]:multiagent—relies on the operation of multiple agents which cooperate in the task of finding the global problem,probabilistic,time-lapse is modeled in discrete manner,operates on the principle of a positive feedback loop—the more agents take part in finding the solution, the higher the quality of the solution will be. This principle holds up to a certain reasonable limit.

### 3.1. Principle of Operation

The first step of the algorithm is placing the ants on the virtual map, which is a fully connected graph, and initialization of the structure representing the pheromone trail. Ants are assigned to random vertices and the initial pheromone level of $\tau_{0}$ is assigned to each graph edge. Each ant initializes its TABU list, which is a structure responsible for tracking vertices that have already been visited by the ant. In the next step, each ant selects an edge that connects its current vertex *i* with another, yet unvisited, vertex *j*. The probability of each transition between vertices in time step *t* is described by the function $P_{ij}\left(t\right)$. After taking a single step, each ant lays down the pheromone trail. The increase in the trail intensity is defined using the function $\Delta \tau_{ij}\left(t,t+1\right)$. The step of selecting the next vertex is repeated until each ant has finished the full graph cycle. Finally, the pheromone trail is also deposited after finishing the cycle (it depends on the ant algorithm version). The algorithm is executed until the halting condition is met—which can be finished given the number of iterations or lack of improvement over subsequent iterations [23]. The general form of the algorithm is summarized in Algorithm 1.

### 3.2. Versions of Ant Systems

Authors of ant systems proposed three variants of the algorithm: ant density, ant quantity, and ant cycle models. The common denominator of the variants is the edge selection step in time *t*. The probability for an ant present in vertex *i* to select an edge connected to vertex *j* is described by the formula:(1)$$P_{ij}\left(t\right)=\left\{\begin{array}{cc} 0 & j\notin J,\\ \frac{{\left[\tau_{ij}\left(t\right)\right]}^{\alpha }{\left[\eta_{ij}\right]}^{\beta }}{\sum_{j\in J}{\left[\tau_{ij}\left(t\right)\right]}^{\alpha }{\left[\eta_{ij}\right]}^{\beta }} & j\in J, \end{array}\right.$$
where:*J* is the set of edges connected to vertex *i*, which has not yet been visited by the ant making the decision,$\tau_{ij}\left(t\right)$ is the density of pheromone trail between vertices *i* and *j* in step *t*,$\eta_{ij}$ is the visibility coefficient of vertex *j* from the perspective of vertex *i*; its value is calculated as an inverse of edge length $\eta_{ij}=\frac{1}{d_{ij}}$,α and β are coefficients that shift importance between pheromone level and visibility.

The general principle of updating the pheromone trail is described by the formula:$$\Delta \tau_{ij}\left(t+1\right)=\rho \tau_{ij}\left(t\right)+\Delta \tau_{ij}\left(t,t+1\right),$$
where the ρ coefficient is responsible for pheromone trail evaporation, which guards against its infinite accumulation. It is essential for its value to be positive, but lower than 1: 0<ρ<1. Value 1−ρ is the remainder of the pheromone trail coefficient.   
**Algorithm 1:** AntSystem**Data**:weighted graph and ant system parameters**Result**:minimal Hamilton cycle
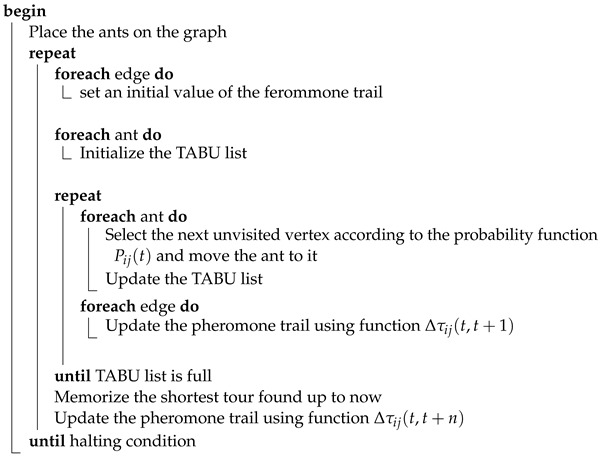


What differentiates the mentioned variations, is the strategy used for calculating the increase in the pheromone level $\Delta \tau_{ij}\left(t,t+1\right)$. In the case of the ant density model, the increment is a constant value for each edge that belongs to the path taken by the ant.
(2)$$\Delta \tau_{ij}\left(t,t+1\right)=\sum_{k=1}^{m}\left\{\begin{array}{cc} Q & \left(i,j\right)\in V_{k},\\ 0 & \mathrm{otherwise}, \end{array}\right.$$
where $V_{k}$ is an edge taken by ant *k* in time step *t*.

When applying the ant quantity variant, the increase in the pheromone level is inversely proportional to the length of the taken edge. This results in further preference towards visibility during edge selection, since shorter edges will gradually become more preferable for ants.
(3)$$\Delta \tau_{ij}\left(t,t+1\right)=\sum_{k=1}^{m}\left\{\begin{array}{cc} \frac{Q}{d_{ij}} & \left(i,j\right)\in V_{k},\\ 0 & \mathrm{otherwise}. \end{array}\right.$$

In the case of ant cycle variation, the pheromone is only updated once, after full algorithm iteration, instead of being incremented after each ant step. Computation of the new pheromone level occurs after *n* steps, where *n* is the length of the routes taken by the ants. The increment $\Delta \tau_{ij}\left(t,t+n\right)$ is calculated for each edge that has been taken by the ant and is inversely proportional to the length of the whole path taken by the ant. The formula describing the increase in pheromone level is described by the Equation (Equation 4). $L_{k}$ denotes the set of edges that belong to the route taken by the ant *k*.
(4)$$\Delta \tau_{ij}\left(t,t+n\right)=\sum_{k=1}^{m}\left\{\begin{array}{cc} \frac{Q}{\left|\right|L^{k}\left|\right|} & \left(i,j\right)\in L_{k},\\ 0 & \mathrm{otherwise}. \end{array}\right.$$

A significant step forward in the realm of ant optimization algorithms is the ant colony system proposed in 1997 [24]. The article proposed three major adjustments to the original algorithm:Extending the edge selection logic with two separate modes of operation: exploration and exploitation. Selection between mentioned strategies happens randomly, based on the value *q* which belongs to the range 〈0,1〉. If its value is less or equal to the hyperparameter $q_{0}$, the ant will select the edge which maximizes the product of visibility and pheromone level. Such behavior exploits the already-known solutions. In the opposite case, the next vertex *s* will be selected according to Formula (Equation 5).Introduction of global pheromone update rule, which is applied after each algorithm iteration. This rule is applied only to the edges which belong to the shortest route that has been found, and is described by the Equations (Equation 6) and (Equation 7).Improvement in the local pheromone update rule, described in Equations (Equation 8) and (Equation 9). Value ρ is the coefficient of evaporation unrelated to α, and $V_{k}$ denotes the edge selected by the ant *k*.
(5)$$s=\left\{\begin{array}{cc} {\mathrm{argmax}}_{u\in J}\left\{\tau_{ij}\eta_{ij}^{\beta }\right\} & q\leq q_{0},\\ \frac{\tau_{ij}\eta_{ij}^{\beta }}{\sum_{j\in J}\tau_{ij}\eta_{ij}^{\beta }} & \mathrm{otherwise}, \end{array}\right.$$
(6)$$\tau_{ij}\left(t+n\right)=\left(1-\alpha \right)\cdot \tau_{ij}\left(t\right)+\alpha \cdot \Delta \tau_{ij}\left(t,t+n\right),$$
(7)$$\Delta \tau_{ij}\left(t,t+n\right)=\left\{\begin{array}{cc} \frac{1}{L_{gb}} & \left(i,j\right)\in L_{gb},\\ 0 & \mathrm{otherwise}. \end{array}\right.$$
(8)$$\tau_{ij}\left(t+1\right)=\left(1-\rho \right)\cdot \tau_{ij}\left(t\right)+\rho \cdot \Delta \tau_{ij}\left(t,t+1\right),$$
(9)$$\Delta \tau_{ij}\left(t,t+1\right)=\sum_{k=1}^{m}\left\{\begin{array}{cc} \tau_{0} & \left(i,j\right)\in V_{k},\\ 0 & \mathrm{otherwise}. \end{array}\right.$$

The authors proved that the proposed ant colony system is capable of achieving results that are at least as good as other heuristics, or even better in major parts of the studied tasks.

As an alternative to the ant colony system, another extension called max-min ant system (MMAS) has been proposed [25]. Its authors recommend the following improvements with respect to the original ant system:After each algorithm iteration, only the ant which traveled the shortest route deposits the pheromone trail. There are two variations of this rule. According to the first one, the shortest route is selected only from the current iteration, and is known as iteration best. Alternatively, one may choose to pick the shortest route from all of the paths traveled across all iterations—such a route is called global best. The formula which describes the pheromone level update is the same as in ant colony system (Equation 7).In order to avoid premature stagnation, which results in all ants selecting the same path, values of the pheromone trail are limited to values which are parameters of the algorithm $\langle \tau_{min},\tau_{max}\rangle$.The pheromone trail is initialized with value $\tau_{max}$. The reason for such a decision is the attempt to encourage the exploration of unknown solutions in the initial iterations of the algorithm. Such a principle lessens the significance of vertex visibility during the edge selection step.

### 3.3. Application of Ant Systems in Steganography

As a result of the advantages of heuristic methods, and hopes for finding a way to optimize two adverse features of steganograms (high capacity and low distortion), attempts in the application of ant and ant colony systems in the task of data hiding in digital images have been made multiple times [26,27,28]. The most important feature, which characterizes the mentioned approaches, is the way of representing the task. In order to apply the ant system metaheuristic, it is obligatory for the input data to be in the form of a graph. The selected method of its construction has a definitive influence on the obtained results and algorithm effectiveness. Another key aspect is the interpretation of the results. There are at least two distinctive directions that experimenters head in. One may treat the shortest route produced by the ants as the end result. Such an approach implies translating the discrete set of taken edges to the pattern in which the data will be hidden. Alternatively, the obtained pheromone trail may be interpreted instead. In such a case, what is obtained is a fuzzy set that represents edges that belong to the best possible route. The analysis of the fuzzy set widens the scope of possible interpretations and has the benefit of higher sensitivity towards selected algorithm variation and values of hyperparameters.

Ant systems could be employed both in spatial [27,28] and frequency-based image steganography [26]. In the article [26], authors proposed a method that relies upon the integer wavelet transform. Experimenters described an algorithm which utilized an ant colony system for data hiding in transform coefficients. Conducted experiments proved the high effectiveness of such an approach [26].

Even though the rest of the mentioned articles describe techniques revolving around spatial image representation, the approach to constructing the graph and interpreting the results is vastly different. In the article [27], images were sectioned into blocks sized 2×2 or 5×5. Each block was interpreted as a separate graph, in which the vertices are pixels and edge distances correspond to an inverse mean square error caused by hiding one bit of information in the specific pixel. The resulting path generated by the ant system indicates which pixels should be used in order to achieve the lowest difference between the transport image and steganogram [27].

An example of a method that utilizes the values of the pheromone trail produced by the ants is [28]. Ants are traveling the graph, which was constructed on the basis of the input bitmap. Its pixels are vertices, and the edge distances are proportional to the difference of intensity between connected pixels. The reasoning behind such representation is the motivation to detect edges and complex image regions, which allows for hiding a higher volume of data while simultaneously preserving lower detectability. After finishing the operation of the ants, a limit value of the pheromone level is selected and each pixel is then assigned to either high complexity or low-complexity image region. As a result, the image is sectioned into two sets of pixels, and in the pixels that belong to one of them, the data will be hidden. With the provided results, it has been proved that this is an effective method. Another benefit of it is the possibility to select and optimize multiple algorithm parameters, such as the method of computing the pheromone limit value, which in turn allows for controlling the quality and capacity tradeoff [28].

## 4. Proposed Technique

The quoted techniques prove the usability and efficiency of the ant system in steganographic algorithms. Moreover, a review of published literature suggests that the topic and exploration of possible solutions is not yet exhausted, and gives hope that finding methods that are more efficient is still plausible. The section presents fundamental assumptions under which the proposed methods for optimizing the trade-off between high capacity and low distortion were developed and validated.

### 4.1. Assumptions

The first fundamental choice that has been made during the development of the proposed technique was deciding upon the utilization of image sections that were characterized by the highest complexity, such as edges and complex textures. Existing literature in the domain of steganography suggests manipulating such regions produces better results, both in terms of capacity and detectability. An example of a widely acknowledged technique that also relies on such an assumption is Pixel Value Differencing described in Section 2.2, which is based on the dependence between the number of modified bits and the measure capturing the difference with neighboring pixels. Another example that supports such a hypothesis can be found in methods that operate on the frequency domain of the image [16].

Based on the obtained sections of the image which are characterized by different levels of complexity, one can conduct the process of data hiding in such a manner that favors complex pixel regions. An accurate analogy to the proposed method can be drawn from the Variable Significant Bit method, which is an extension of the Least Significant Bit technique. In the approach described below, the membership rate to an image complex region dictates the number of bits that will be replaced by the secret message in the process of data hiding. The main reason for representing the complex regions as fuzzy sets, in opposition to the approach described in [28], is its greater potential for achieving greater steganogram capacity. When settling for binary image partitioning, the solution is bound to make use of two distinct numbers of bits that will be substituted. In the particular case of the mentioned article, the number of used bits in the low complexity region was equal to zero. A downside of such an approach is the inability to differentiate and therefore efficiently exploit regions of medium complexity. Placing the complexity on a continuous scale allows for more efficient utilization of the whole image space of varying complexity. The approach grants not only the utilization of a larger number of pixels but also the obtained steganogram may be characterized by lower detectability since the regions which contain large amounts of confidential data are not separated by a sharp edge from the remaining sections of the image.

### 4.2. Employing Ant Optimization

The stage of the steganographic process that was regarded as a step that could potentially benefit the most from the ant system metaheuristic was detecting and assessing the complexity of image regions. In the context of the proposed method, the following subject is key to the final results, as it affects whether the data will be hidden in a manner that impedes message detection and maximizes the image capacity. The described technique relies on a specific method of transformation from a transport image bitmap into a graph, conducting a preset number of ant system iterations, reading and interpreting the deposited pheromone trail. The stage of pheromone trail interpretation is tightly linked to the implemented method of bitmap transformation, and its goal is obtaining a masking matrix whose dimensions correspond to the input image size. Values of the matrix coefficients are later used during the calculation of the number of bits that will be substituted with secret information in each pixel. Principles of operation are summarized in the following steps:Determine graph representation of the input bitmap.Execute *n* iterations of selected ant system variant with a given set of parameters.Read obtained pheromone trail.Based on the trail, calculate values of the masking matrix $K_{xy}$ of dimensions equal to an input image.In each pixel substitute $k_{ij}$ least significant bits with bits of secret message.

The above process can be described by the diagram on Figure 1 and Algorithms 2 and 3.
**Algorithm 2:** EmbedDataInInputImage**Data**:*I* – structure representing RGB pixels of an input image*D* bits of data to be hidden**Result**:*S* – structure representing RGB pixels the steganogram
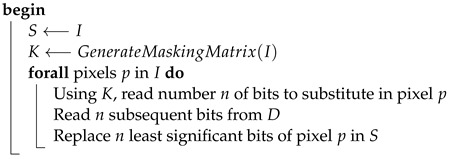


**Algorithm 3:** GenerateMaskingMatrix**Data**:*I* – structure representing RGB pixels of an input image**Result**:*K* – matrix equal in dimensions to *I*which values control how many bits should be substituted

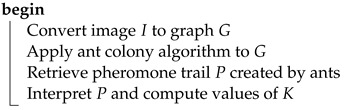



Extraction of the message embedded in the image is analogous to the process of hiding. To read the message, the masking matrix needs to be generated again and used during the sequential reading of the established number of bits from each pixel.The summary of this step is presented in the diagram on Figure 1 (see comment) and Algorithm 4.
**Algorithm 4:** ExtractDataFromStegoImage**Data**:*I* – structure representing RGB pixels of an input image*S* structure representing RGB pixels of a steganogram**Result**:*D* – bits of hidden information
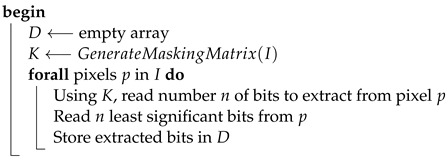


Utilization of an ant system has additional benefits in terms of security. The obtained pheromone trail, therefore the masking matrix, is dependent on, and very sensitive to, changes in selected hyperparameters. Thanks to that, in the hypothetical case of a steganogram being intercepted by a third party who acknowledges the mere fact of private communication taking place, capturing its content will be significantly impeded. This means that the parameters of the ant system represent a form of a cryptographic key.

### 4.3. Graph Image Representation and Pheromone Trail Interpretation

As previously stated, the task of transforming the input bitmap into a graph is an essential step to utilizing the ant system in order to optimize the steganographic process. The characteristics of obtained graph, such as the number of vertices and edges will have a direct influence on algorithms performance and efficiency.

An important aspect of the technique is interpretation of the pheromone trail. The deposited trail consists of numeric values which are assigned to each edge of the graph. Consequently, its interpretation cannot be detached from the method by which it was created in the first place, and therefore it was given a certain meaning. In reference to the above statement, proposed strategies for constructing the graph will be presented in parallel to methods of computing the making of a matrix based on the pheromone trail.

### 4.4. Vertex-Based Method

In the first proposed method, graph (V,E) is constructed starting from the vertices. Each pixel of the input image corresponds to a vertex, which means that the size of the set of vertices *V* is equinumerous to the set of pixels. Pixel neighborhood can be either understood as a 4-connected or 8-connected manner, which is depicted in Figure 2.

Length of edge $E_{ij}$ connecting pixels represented by vertices $V_{i}$ and $V_{j}$ is calculated on the bases of the difference between pixel intensity of all *RGB* channels, which is expressed by the formula:(10)$$\Delta p_{ij}=\frac{{\left(p_{i}^{r}-p_{j}^{r}\right)}^{2}+{\left(p_{i}^{g}-p_{j}^{g}\right)}^{2}+{\left(p_{i}^{b}-p_{j}^{b}\right)}^{2}}{255^{2}\cdot 3},$$
where $p_{i}$ denotes a specific pixel, and $p_{i}^{r}$, $p_{i}^{g}$, $p_{i}^{b}$ are its components. Such a strategy of distance calculation encourages ants to select, and therefore deposit pheromone trail, on edges connecting alike pixels, if the distance will be inversely proportional to the intensity difference, or differential otherwise.

A graph constructed using such a method, described by Algorithm 5, is not fully connected. That means that the task it represents is not equivalent to the traveling salesman problem (TSP). Another difficulty that sprouts from the above method is the numerosity of vertices. Searching for Hamiltonian cycles in the graph of w×h vertices is at the very least ineffective, since the product of width *w* and height *h* reaches high values even for small images.
**Algorithm 5:** ConvertImageToGraphVertexBased**Data**:*I* - structure representing RGB pixels of an input image**Result**:*G* - graph representing the input image
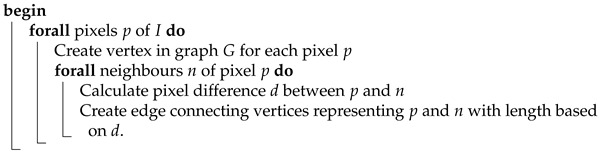


As a result, in order to employ the following method, some changes have been made to the way the ant system was used and configured. The most significant of them involves desisting from making each ant finish the full cycle. Instead, the number of executed steps in each iteration is another parameter of the algorithm. Additionally, in order to improve performance on high-resolution images, another feature was implemented, which enables scaling down the image before graph construction and upscaling the resulting masking matrix back to initial dimensions w×h.

As another implied consequence, small adjustments had to be made to the pheromone update rule. Since the constructed graph is not fully connected, there is a chance for an ant to get stuck in a position where it cannot make the next step, even though it has not yet traveled the defined number of edges. The route of the ant will be shorter than the routes of other ants and therefore will be unfairly favored. All systems which calculate the pheromone update based on the whole route length, such as the ant cycle model of the classic ant system, ant colony system, and max-min ant system, are vulnerable to that pitfall. In order to account for it, a route length coefficient which scales the length to target the number of steps was introduced. For example, the adjusted pheromone update Formula (Equation 4) became:(11)$$\Delta \tau_{ij}\left(t,t+n\right)=\sum_{k=1}^{m}\left\{\begin{array}{cc} \frac{Q}{|L^{k}|\cdot \frac{|L^{k}|}{N}} & \left(i,j\right)\in L_{k},\\ 0 & otherwise, \end{array}\right.$$
where $|L^{k}|$ means number of steps in route $L^{k}$ and *N* denotes the target number of steps.

Since the masking matrix must be of the same dimensions as the input image, the coefficient at position x,y corresponds to pixel $p_{ij}$. In order to compute the values of the matrix elements $K_{xy}$, which dictate the number of bits to be substituted by the bits of the secret message in each pixel, it is required to take into account all of the edges that are connected to the vertex the pixel $p_{ij}$ is related to. This process is explained by Algorithm 6.
**Algorithm 6:** ConvertPheromoneToMaskingMatrixVertexBased**Data**:*I* - structure representing RGB pixels of an input image*P* - pheromone trail deposited by ants**Result**:*K* - matrix equal in dimensions to *I*which values control how many bits should be substituted
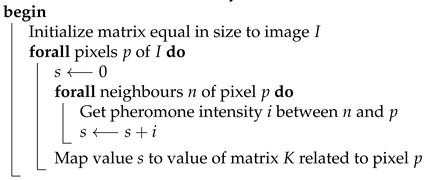


A dilemma that arose during the development of the method was the nature of the relation between a numeric difference between neighbor pixels $\Delta p_{ij}$ and the length of graph edges. Initial experiments showed higher effectiveness of the configuration in which edge length is proportional to pixel difference. This in turn implies that laying the pheromone trail will be more frequent and intense on edges that connect alike pixels. In order to account for that, the number of utilized pixel bits is inversely proportional to pheromone trail intensity.

In accordance to the above, the value $K_{xy}$ is described by the following formula:(12)$$K_{xy}=255\cdot \left(1-\frac{\sum_{j\in A_{i}}\tau_{ij}}{|A_{i}|}\right),$$
where $A_{i}$ means the set of neighbor pixels to pixel *i*.

Figure 3 presents the input image, which was overlaid with lengths of edges linked to specific pixels.

### 4.5. Edge-Based Method

The second proposed approach is constructing the graph starting from the edges. The first step is to partition the image into a specified number of segments, which is significantly lower than the number of pixels. Each segment is represented by one edge in a graph. Their lengths are dependent on the variance of every pixel that belongs to the specific segment. As in the case of constructing the graph starting from the vertices, edge length can be either directly or inversely proportional to the selected measure, in this case, variance. A justification for selecting variance as the measure describing each segment is its ability to capture deviations between pixels belonging to it. Segments that are in high-complexity areas will have higher variance.

Subsequently, out of the obtained edges, a fully connected graph is constructed. In order to make it possible, it is required to satisfy the condition:(13)$$\left|E\right|=\frac{\left|V|\cdot (|V|-1\right)}{2}.$$

Due to the above fact, implemented Algorithm 7 takes as a parameter only the number of vertices |V|, and the number of edges |E|, and therefore a number of image segments is calculated automatically.
**Algorithm 7:** ConvertImageToGraphEdgeBased**Data**:*I* - structure representing RGB pixels of an input image*v* - target number of graph vertices**Result**:*G* - graph representing the input image
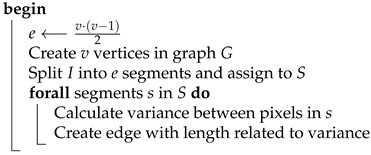


Afterwards, the graph is utilized by the ant system. The task that is set upon virtual ants can be interpreted as follows: out of |E| edges, select ones that will allow for hiding information in the most complex regions of the image. Such a task is equivalent to the traveling salesman problem, and therefore no modifications to the algorithm need to be made.

The resulting pheromone trail assigns a numeric value to each edge of the graph. In order to calculate the masking matrix, the pheromone value of specific edge $E_{i}$ needs to be overlaid over pixels that belong to the segment it represents. Since each pixel is assigned to exactly one segment, computing the masking matrix does not impose any further difficulties. The conversion process is described by Algorithm 8.   
**Algorithm 8:** ConvertPheromoneToMaskingMatrixEdgeBased**Data**:*I* - structure representing RGB pixels of an input image*P* - pheromone trail deposited by ants**Result**:*K* - matrix equal in dimensions to *I*which values control how many bits should be substituted
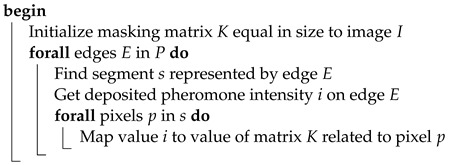


Implementation of the above method cannot be separated from the task of image segmentation. Selection of the segmentation strategy is not obvious, since there are numerous ways in which such division can be made. As a result, it was decided to implement three different segmentation methods and compare their results. The implemented methods are described in the following list:Image segmentation into a specified number of non-overlapping rectangles in axes *x* and *y*. It is necessary to select such a number of splits $S_{x}$ and $S_{y}$ in axes *x* and *y* that their product is equal to |E|. Otherwise, constructing the graph will not be possible.An advantage of this method is its simplicity and intuitiveness, yet it is not without drawbacks. In the case when bitmap dimensions are not divisible by $S_{x}$ and $S_{y}$, it is required to enlarge segments in the last row and column. Another disadvantage is sharp edges between segments, which can manifest in higher pronunciation of the quality difference between neighboring segments. Figure 4 presents segmentation visualization accomplished using this method.Image segmentation can also be achieved with the use of clustering algorithms. An example of such an algorithm is *k*-means. Its application allows for partitioning input data into a given number *k* of groups in such a way that the distance between each group member is minimized [29]. The algorithm allows for the clustering of multidimensional data, but in this case each pixel was only characterized by its variance with surrounding neighbors. Since such a grouping does not take into account spatial information, pixels that belong to the same group are not necessarily direct neighbors. Figure 5 presents examples of groupings into a given number of clusters.The last used segmentation method is partitioning images into superpixels. Superpixels are groups of pixels that are characterized by similar components in the color space. Applied algorithm *SLIC* relies on *CIELAB* color space [30]. The popularity and adoption of superpixels in the computer vision domain is constantly increasing since it allows for capturing the most important characteristics of an image with a significant reduction in its size. This in turn results in shorter execution time of numerous image processing algorithms. Their uses can be found in object detection and identification tasks. In opposition to *k*-means algorithm, obtained groups of pixels are adjacent. Figure 6 presents segmentation visualization using the described method.

## 5. Experiments and Results

In order to assess the effectiveness of the proposed techniques, a series of experiments were conducted. Their goal was to verify fundamental assumptions, such as the validity of selecting complex image regions, testing the efficacy of different methods of constructing graphs and pheromone trail interpretation, quantitative assessment of steganogram quality degradation, and subjective perceivability of the introduced changes. Quantitative evaluation of image distortions allowed for objective result comparison to established techniques and answering the question of proposed algorithms’ utility.

### 5.1. Research Method

Verification of the proposed method consisted of conducting the process of embedding data in bitmaps commonly used in the computer vision domain.

#### 5.1.1. Images

Despite the existence of alternative sets of test images consisting of synthetically generated images that aim at highlighting particular features and image distortions [31], a collection of real pictures that better reflect the practical uses of steganography were used instead. Their selection was mostly based on the number of references in related articles and current tendencies in the use of public images [32,33]. Finally, it was decided that the experiments will be conducted on the following images: *Mandrill* (also known as *Baboon*), *Airplane* (also known as *F-16* or *Jet*), *House*, and *Peppers*, all of which are published by *University of Southern California* [34]. All of them are presented in Figure 7.

#### 5.1.2. Data

Data that were embedded in test images were in the form of *ASCII* text, but after minor adaptations, the algorithm could be easily used for embedding arbitrary binary sequences. During the experiments the used text was automatically generated *Lorem Ipsum* of size 625 kB, but during most of the trials only a subset of it was used. A feature of the implemented algorithm was the possibility to scale the generated masking matrix in such a manner, that embedding the text would result in specified target capacity. This allowed for assessing the quality degradation in relation to the volume of hidden data, and simplified result comparison with other techniques present in the literature.

#### 5.1.3. Algorithm Parameters

During the conducted experiments the influence of particular methods and their parameters related to each step of the algorithm were assessed. Compared values of parameters referred to:Graph construction and pheromone trail visualization.Vertex-based method. The only optional argument that parametrizes the method is the scaling factor $s_{0}$ used for downscaling the input image and upscaling the masking matrix. Its sole purpose was to enable the construction of reasonably sized graphs for high-resolution images. Its value is contained within the [0,1] range. Its default value is 1 and results in the construction of graphs with a number of vertices equal to w·h.Edge-based method. It is primarily parametrized by the selected segmentation algorithm and the target number of segments $N_{s}$, which is closely related to the number of graph edges. During experiments following algorithms were used: rectangular segmentation, *k*-means, and *SLIC* used for superpixel segmentation.Determining the pheromone trail values by various kinds of ant systems. Parameters common to each of the system variations are as follows:number of ants *A*, which by default is equal to number of graph vertices |V|,number of algorithm cycles *C*,number of steps executed by each ant in each iteration *S*, for graphs constructed based on the segmentation method it is equal to number of vertices |V|,bias towards pheromone trail level α,bias towards vertex visibility β,initial pheromone level $\tau_{0}$,pheromone evaporation rate ρ.The following variations of the ant system were examined, along with additional parameters linked to their mode of operation:Ant density model,Ant quantity model,Ant cycle model,Ant colony system, which introduces exploitation probability $q_{0}$ and fixes α=1,Max-Min ant system, which introduces bounds for value of pheromone trail $\left[\tau_{min},\tau_{max}\right]$. Their values are computed based on estimation concerning the target cycle length. Additionally, the initial pheromone value is$\tau_{0}=\tau_{max}$.

#### 5.1.4. Image Quality Measurement

In order to efficiently evaluate the quality of the steganograms, it was decided to utilize several image quality metrics. The list of applied metrics can be partitioned into two categories: objective and subjective. Objective metrics serve the purpose of comparing characteristic values between two signals. In the case of subjective metrics, their task also consists of computing numeric value which represents the image, however, the main focus is placed upon a correlation of its values with the subjective impression of the human vision system. The list of used metrics is as follows:Mean Square Error (MSE)Is one of the simplest metrics that can be used to measure differences between two images. Its values belong to a set of non-negative real numbers, and values close to zero denote lower distortion. Its main advantage is its ease of implementation and possible optimization. One of its disadvantages is its low correlation with perceived differences between two images and unawareness of the ratio between signal and noise. Its value is expressed by the formula:
(14)$$\mathrm{MSE}=\frac{1}{n}\sum_{i=1}^{n}\left(X_{i}-\bar{X}\right),$$
where:*n* – number of observations,$X_{i}$ – signal value,$\bar{X}$ – mean signal value,Peak Signal Noise Ratio – PSNR.The peak signal-to-noise ratio is an improvement over the mean square error since this metric is sensitive to the maximum value of the signal. This implies that the same value of *PSNR* translates to differences proportional to the amount of information. For example, in the case of *MSE* the same value will translate to different perceived distortion in images that use 8 and 24 bits per channel. The peak signal-to-noise ratio solved the above issue. Because of the large value range, it utilizes a logarithmic scale. Values that are common in the area of 8-bit image analysis belong to the range [30 dB, 50 dB]. Higher values are linked to lower degradation.
(15)$$\mathrm{PSNR}=10\cdot log_{10}\frac{{\mathrm{MAX}}^{2}}{\mathrm{MSE}},$$
where:MAX – maximum signal value,MSE – mean square error.Structural Similarity Index – SSIM.The most complex metric that was used during evaluation is *SSIM*. Its main goal is to capture the complexity and features of the human vision system. It relies on the hypothesis asserting the importance of image structure in relation to perceived luminance and contrast [35,36]. Its value is commonly normalized to the range [0,1], where 1 signifies image identity.

### 5.2. Results

Conducted experiments have been performed with the analysis of the vertex-based method graph construction.

#### 5.2.1. Vertex-Based Method

The first factor that was analyzed was the influence of the proportionality between lengths of graph edges on the difference between pixels. As explained in Section 4.3, if the edge lengths are proportional to pixel differences, the ant system will be converging towards depositing larger amounts of pheromone on low complexity regions. In order to achieve the desired result, which is the masking matrix that exposes areas of high complexity, its values need to be inverted. Alternatively, if the difference between pixels will be inversely proportional to the distances, the resulting pheromone trail will highlight more complex regions.

Visualization of the graph construction process and the resulting masking matrices are presented in Figure 8. The first column contains images demonstrating bitmap to graph conversion, in which each pixel is attributed luminance inversely proportional to linked edge length (areas more “attractive” to the ants are represented by brighter pixels). The second column contains masking matrices which dictate the number of bits that will be substituted with bits of the secret message. The brighter the pixel is, the more bits will be replaced. The last columns contain the same masking matrices after adjusting their values to accommodate the same steganogram capacity.

Based on the Figure 8b,c several conclusions can be drawn. Both methods detect edges and complex regions correctly, but after adjusting the matrices to the same capacity, a conclusion that recognizes the superiority of the approach involving distances proportional to pixel difference can be made. Scaled and inverted masking matrix Figure 8f guarantees more even information distribution while still favoring the complex regions.

The next parameter which was studied was the variation of the ant system and its parameters. Since the approached problem was not equivalent to the traveling salesman problem, it was also necessary to determine the value of the *S* parameter, which denotes the number of steps each ant takes in each algorithm iteration.

Since the remaining images from the *USC* dataset have dimensions of 512×512, the next step in the study was assessing the influence of the scaling factor *S* which downscales the image during graph construction. During these trials, the ant cycle model was used with the following parameters: A= 10,000, S=100, α=β=1, ρ=0.8, and a target capacity of 250 kB. Figure 9 presents masking matrices generated with $s_{0}\in \left\{0.25,0.5,0.75\right\}$. Table 1 contains quality measurements of the steganograms.

On the basis of the results, it is safe to assume that scaling the image does not have a significant negative impact on the steganogram quality. This is a desired feature since computations conducted on smaller graphs take less processing power, and thus time.

Finally, the results were compared across different test images. Obtained results are presented in Table 2, and the steganograms are visible in Figure 10 (sample differences within the images are pointed out). Algorithm parameters were preserved from the previous experiment.

#### 5.2.2. Edge-Based Method

The next step in the experiments was the examination of the method involving graph construction starting from edges. In the case of such a method, the image is segmented into a target number of regions, which relate to edges of the fully connected graph. Since the graph is fully connected, the number of segments must obey the equation expressing the dependency between a number of edges |E| and vertices |V|: $\left|E\right|=\frac{\left|V|\cdot (|V|-1\right)}{2}$. The traveling salesman problem expressed on such a graph can be interpreted as the task of finding *n* segments which constitute the shortest cycle. The edges’ lengths of such a graph are dependent on the variance within the segment.

Similarly to the vertex-based method, firstly the relation between edge length and distance metric was assessed.

Figure 11 presents a visualization of the constructed graph and obtained pheromone trails. Compared images were partitioned using 190 superpixels (|V|=20). Based on the steganograms, the downside of the inversely proportional edge-length-to-segment variance can easily be observed (Figure 11a–c).

When solving *TSP*, the ultimate result is selection of only |V| out of $\frac{\left|V|\cdot (|V|-1\right)}{2}$ edges. This implies that once the ant system converges, the pheromone trail on all edges except the ones belonging to the shortest route will be close to zero. In such a case, only a small image subset will be used for actual data hiding, which negatively affects the steganogram capacity. The chart in Figure 12 presents the ratio between |V| to the number of edges of the fully connected graph.

The above issue can be solved in at least two ways. The first one is a precise adjustment of the parameters in a way that will promote exploration (e.g., high value of $q_{0}$) or more greedy behavior of the ants (by adjusting parameter α). As a result, a higher number of edges will be visited at the cost of a higher risk of suboptimal solutions. Alternatively, the task represented by the graph can be inverted. If edge lengths are proportional to segment variance, ants will converge to finding |V| edges in which secret information should not be hidden since related segments are not complex enough. The resulting pheromone trail will promote $\left|V\right|-\frac{\left|V|\cdot (|V|-1\right)}{2}$ segments in which information should be deposited.

Since the approached task was analogous to the traveling salesman problem, during the experiments it was appropriate to utilize optimal parameter values for each ant system variant, which were determined during testing of the algorithm implementation validity. Methods of their computation were based on the information provided in related research papers.

In the method that involved image segmentation, an aspect of significant importance is the target number of pixel groups. Too low a number may result in segments that mix homogenous and complex regions, thus producing suboptimal bit substitutions. Too high a number, aside from incurring higher computational intensity, may lead to the creation of segments in size not sufficient enough to truly represent the variance of the image region.

Figure 13 presents the segmentation of each transport image into a varying number of segments with pheromone trail overlay. Table 3 contains values of quality metrics of the generated steganograms for an input image of size 512×512 and target capacity 25 kB.

Each of the segmentation methods produced similar results, but it is also worth mentioning that rectangular and superpixel segmentation require exceeding a specified threshold, above which acceptable results are achieved. In relation to the compared results, it was decided to segment images into 20 superpixels.

The last step of edge-based method analysis was comparing the results to the rest of the test images. Results are presented in Table 4 and steganograms can be found in Figure 14 (sample differences within the images are pointed out).

### 5.3. Subjective Evaluation

Generated steganograms fulfill the expectations fully. For images in which ∼32% capacity was substituted for the hidden message, distortions are practically imperceptible. Artifacts became apparent once the hidden capacity was 500 kB, which is roughly 64% of their volume. Changes were more apparent in images generated using rectangle and superpixel segmentation, due to the borders between groups of pixels. The vertex-based method of graph construction guarantees smoother transitions between regions that utilize a higher number of pixel bits.

It is also worth pointing out that subjective impressions mostly correlate to the values of used quality metrics. Additionally, the Peak Signal to Noise Ratio range, 30–50 dB, also corresponds to perceived quality. For acceptable steganograms, which utilized ∼32% of image capacity, *PSNR* the value was close to 40 dB. When using ∼64% of the image bits, *PSNR* the value dropped to an unacceptable level close to 20 dB.

Another observation can be derived from the comparison of results among different test images, especially *Baboon* and *Airplane*. Even after embedding 500 kB, distortions were more noticeable in the *Airplane* image. Such a conclusion may be justified by a much greater number of details, textures, and overall complexity of the *Baboon* image.

### 5.4. Result Comparison against Other Methods

The last step of the effectiveness evaluation of the proposed techniques was the result comparison against metric values published in related articles. Published studies allowed for result comparison between the following methods: *Least Significant Bit* [37], *Pixel Value Differencing* [37], *Particle Swarm Optimization-Integer Wavelet Transform* [18] and method of complex region detection using ant system [28] discussed in Section 3.3.

As a result of the fact that the above articles published their results for different steganogram capacities, it was decided to place specific comparisons in separate tables. Formats of the results were inconsistent. Some of the articles used capacity expressed as a percentage of substituted bits, number of bits per pixel (*bpp*), or absolute number of bytes. However, since the test images were known, it was possible to convert the results to a common format.

Comparison between specific articles can be found in Table 5, Table 6, Table 7 and Table 8. In columns containing metric values, values in bold denote the lowest image distortion.

Based on the Table 5, comparing the results between *LSB* and techniques proposed in this article, it is difficult to conclude the superiority of one approach over another. For each of the test images values of *SSIM* were preferred in favor of *LSB* method. Values of *PSNR* were comparable for all approaches.

Such results may be explained by the nature of *SSIM* metric, which focuses on differences in image structure. The Least Significant Bit method introduces homogenous noise on an equal level to the processed image, which affects its structure to a small extent, and therefore produces a small change in the metric value.

Result comparison between *PVD* and proposed techniques (Table 6) clearly favors the vertex-based method. *PSNR* values significantly surpassed results published in article [37]. Values of *SSIM* were similar, but in two out of three images, they were advantageous for the described method.

Comparison of metric values with the *PSO-IWT* technique described in [18] also suggests the preferability of the proposed methods (Table 7). Due to missing *SSIM* values, it was only possible to make a comparison against the *PSNR* metric, whose values are higher by roughly a degree of magnitude.

Similarly to previous comparisons, Table 8 presents results between proposed methods and the method which was ideologically closest to it, since it also involved utilizing an ant system for complex image region detection. Values of *PSNR* and *SSIM* metrics advocate for superior efficiency of the proposed methods in terms of capacity for degradation tradeoff. By contrast to previous juxtapositions, slightly better results were obtained using the edge-based method.

### 5.5. Detectability and Resilience against Steganalysis

Another aspect that is crucial to all steganographic techniques, which greatly affects their applicability, is the difficulty to identify the existence of the secret message. Unless the hidden data are secured with cryptographic keys, detecting the presence of the message may lead to disclosing and exposing all of the information conveyed.

One of the steganalytic methods which is proven to be highly effective against digital image steganography is RS analysis [38]. Its main focus is detecting and estimating the size of the hidden message in digital images subjected to *LSB* manipulation. Since the proposed methods can be seen as derivatives of *LSB* and *PVD*, it seems logical to apply similar steganalytic techniques to assess their resilience to attacks.

The Table 9 contains the predicted capacity of the message embedded in images obtained using both methods (Figure 10 and Figure 14). Although in all cases the existence of the secret message was disclosed, its size turned out to be most frequently underestimated, in the majority of cases even two-fold. It is also worth noting that even if the message presence was detected, extracting its contents, due to the stochastic nature of the process explained in Section 4.2, would be rather a non-trivial task.

## 6. Result Summary

Conducted experiments can be the basis for further interesting conclusions. Firstly, both proposed methods—constructing graphs starting from vertices or edges—are correct and allow for controlling the process of data hiding. The main criterion which was assessed during method evaluation, was their influence on image degradation in relation to volume of embedded data. Therefore, after selecting and adjusting the parameters of the process, it is safe to conclude that the proposed techniques provide results that are close if not better in relation to methods proposed in [18,28,37].

Both methods described in this article share fundamental assumptions and a general mode of operation. Their differences related to image-to-graph conversion are evident, however, results obtained by their operation are similar in regard to metrics described in Section 5.1.4. The above fact can be interpreted as successful verification of the fundamental assumptions, such as the validity of choosing complex regions, which were described in Section 4.

The parameter which had the most significant influence on the obtained results was the relation between graph edge length and differences between pixels or within segments. The experiments showed an undeniable advantage of the scenario when edge lengths are proportional to pixel differences or segment variance. Implementing such an approach leads to more intensive deposition of pheromone in homogenous image areas, and enforces calculation of an inverted masking matrix during data embedding and extraction. Numerous root causes can be inferred here. For example, one reason for such results could be a smaller number of complex regions in relation to homogeneous ones. Alternatively, complex regions of images may be often separated by low complexity areas. In both cases, such characteristics hinder the virtual ants’ exploration of the whole image, which results in the pheromone trail being too condensed, such as in Figure 11c.

An interesting feature that differentiates the proposed methods is the nature of the task represented by the constructed graph. In the case of the vertex-based method, the task being solved by virtual ants is not analogous to the traveling salesman problem, and requires specific adaptations described in Section 4.4. This fact made the experimentation and evaluation much harder, since the heuristics of parameter selection proved to be ineffective. Additionally, this method is highly sensitive to the kind of ant system and turned out to be inapplicable for the ant colony system and max-min ant system. Such results may be explained by their strong bias towards the single best route, which may not fit the expected solution. The technique in which the graph was constructed starting from the edges was free from the above issues. The represented task was analogous to the traveling salesman problem, and thus it was possible to utilize suggestions and experimentally determined heuristics for parameter selection.

Since both methods generated similar results, it is possible to select the method which is less computationally intensive. During the experimental process, the least taxing methods were ones based on the rectangle and superpixel segmentation. Due to regular edges separating rectangular groups of pixels in which the same number of pixels was substituted it is preferable to use superpixel segmentation.

## 7. Conclusions

The main goal of the paper was to study the feasibility of applying ant optimization algorithms in steganography, proposing possible approaches, and evaluating their effectiveness. The article presents two techniques utilizing ant systems in image-based steganography. The primary focus of the experiments was the aspect of image degradation in relation to data capacity, which is key for confidential communication.

Proposed and implemented methods proved to be effective and allowed for embedding large amounts of data with a relatively low impact on image quality. The influence of ant system variations and parameters was also studied, and the final effectiveness was also evaluated. Presented experiment results include both elements of subjective assessment and values of quality metrics which allow for unbiased comparisons. Results obtained via experiments on reference images were then juxtaposed with related articles, and the competitiveness of described methods was concluded.

Out of all possible potential developments, the idea which could be further explored is the usage of different, or larger numbers of image preprocessing algorithms. Experiments concerning the method based on vertices proved that scaling the input image during graph construction does not have a significant detrimental effect on steganogram quality, but allows for undeniable performance gains. Moreover, it cannot be ruled out that more sophisticated image transformations could further improve the characteristics of the proposed techniques. Another topic worth researching would be studying and assessing steganogram vulnerability to distortions, manipulations, and steganographic attacks. 

## Figures and Tables

**Figure 1 entropy-25-00963-f001:**
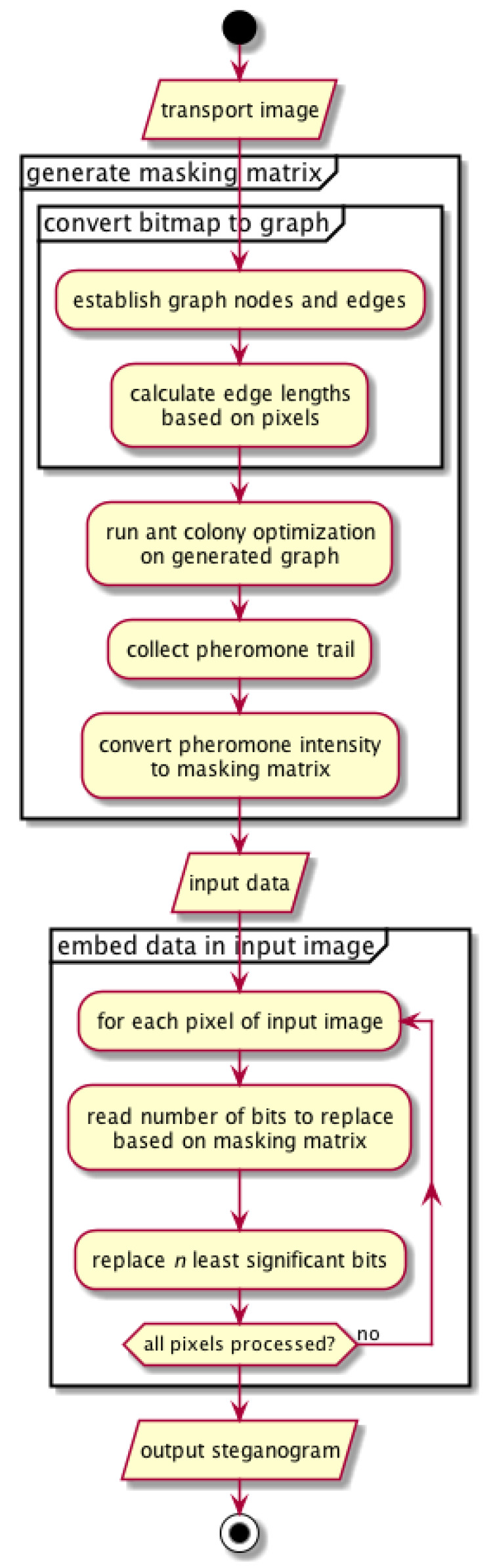
Process of data hiding (the process of data extraction is similar; only at the point of data extraction there is an image with hidden data that will appear in the output).

**Figure 2 entropy-25-00963-f002:**
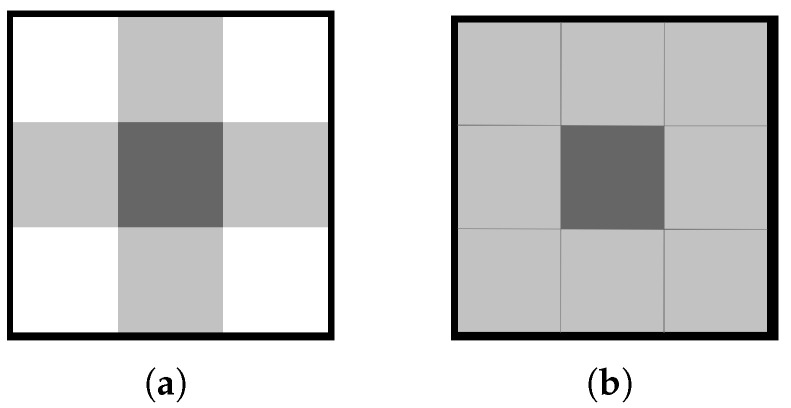
Types of pixel connectivity: (**a**) 4-connected neighborhood; (**b**) 8-connected neighborhood.

**Figure 3 entropy-25-00963-f003:**
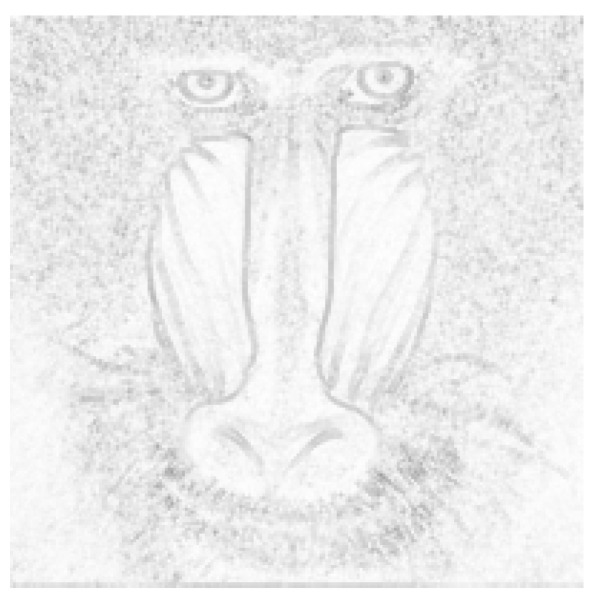
Visualization of the graph constructed using the vertex-based method. Brighter pixels correspond to vertices that are connected to shorter edges.

**Figure 4 entropy-25-00963-f004:**
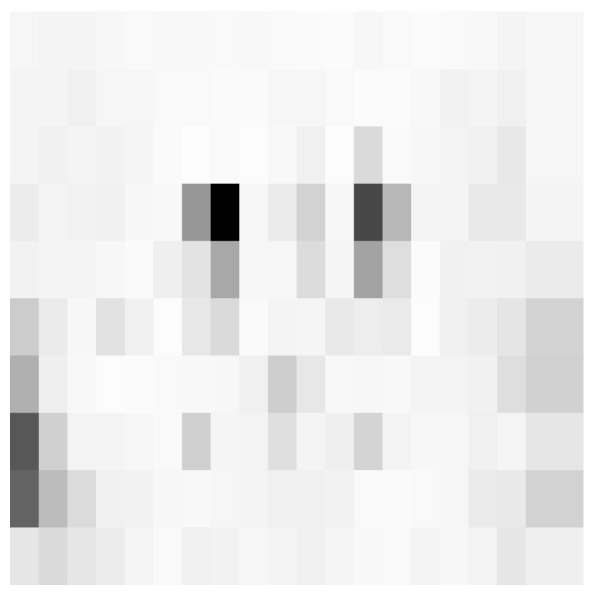
Graph construction visualization using the rectangle segmentation method.

**Figure 5 entropy-25-00963-f005:**
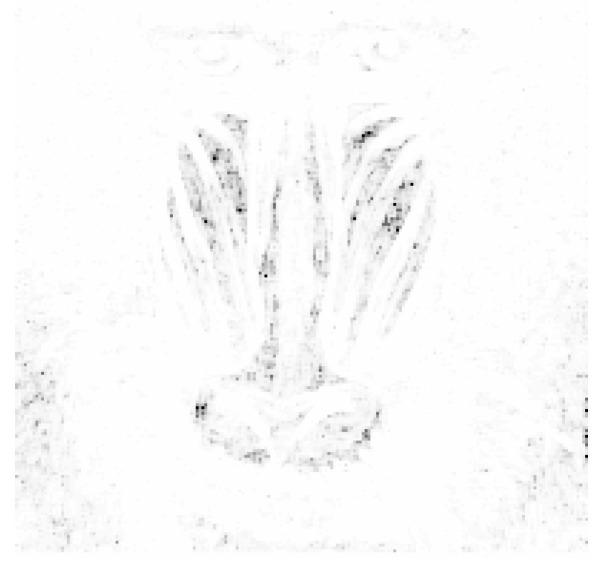
Graph construction visualization using *k*-means algorithm.

**Figure 6 entropy-25-00963-f006:**
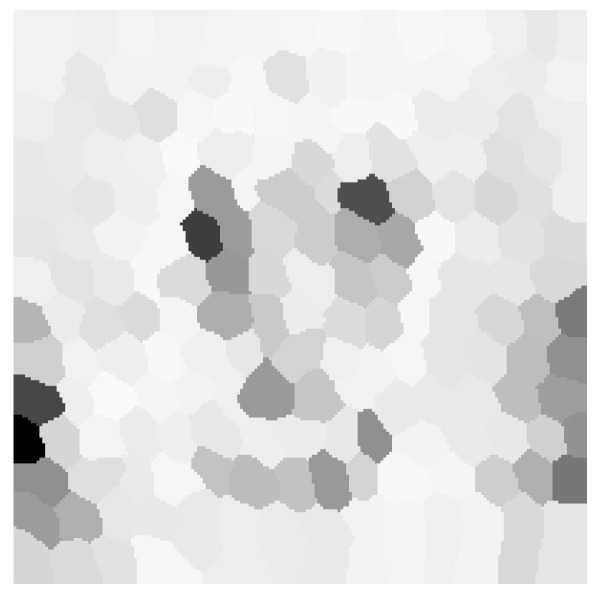
Graph construction visualization using superpixel segmentation.

**Figure 7 entropy-25-00963-f007:**
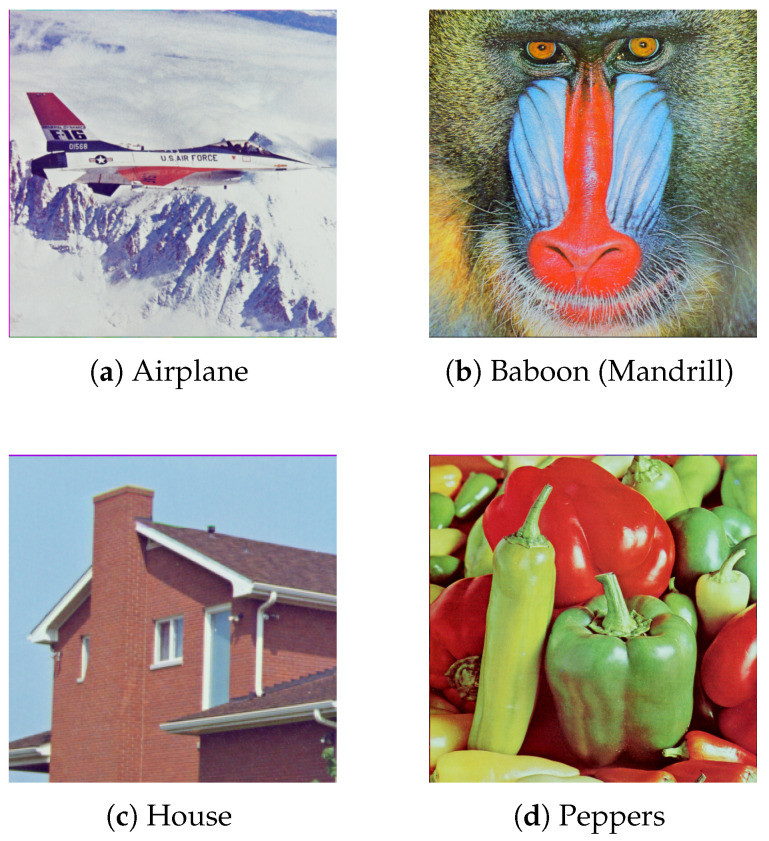
Images used in experiments.

**Figure 8 entropy-25-00963-f008:**
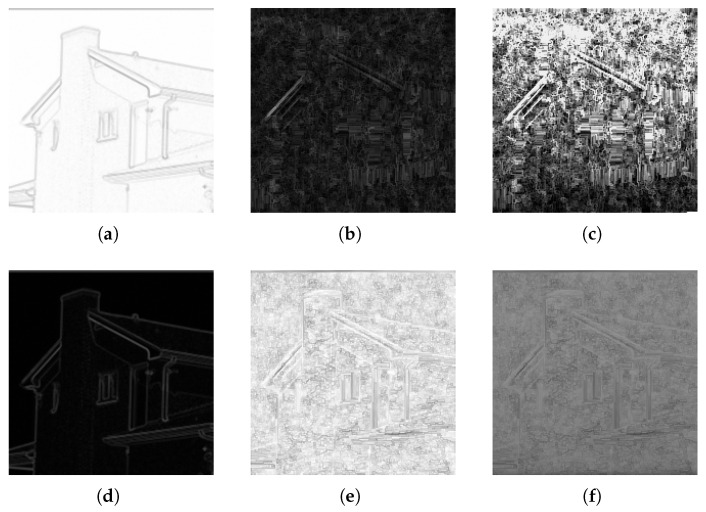
Comparison of graph conversion visualizations and obtained masking matrices. The first row (**a**–**c**) presents images related to the process with distances inversely proportional to the pixel difference. The second row (**d**–**f**) presents the results if distances are proportional to differences. The first column contains image-to-graph conversion visualizations, the second presets generated masking matrices, and the last one contains the matrices after adjusting them to the same steganogram capacity.

**Figure 9 entropy-25-00963-f009:**
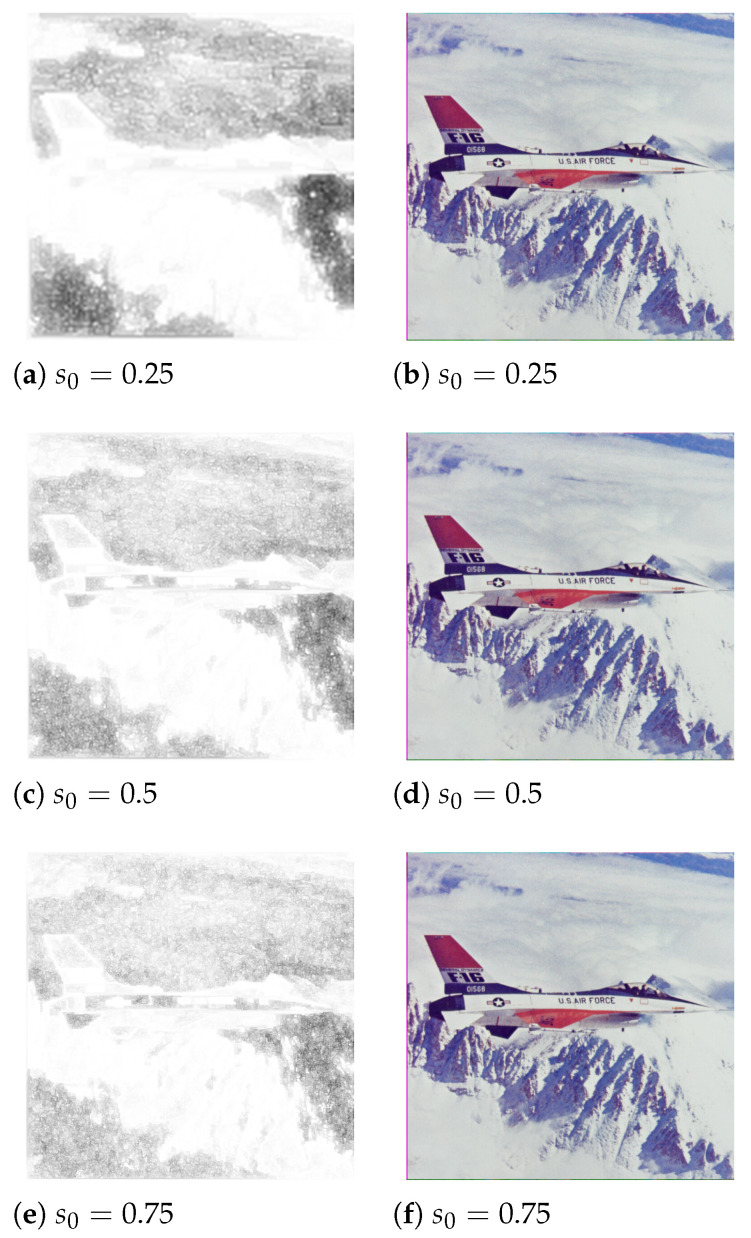
Sample masking matrices and steganograms generated with different values of scaling factor $s_{0}$ ((**a**,**c**,**e**)—masking matrices, (**b**,**d**,**f**)—steganograms).

**Figure 10 entropy-25-00963-f010:**
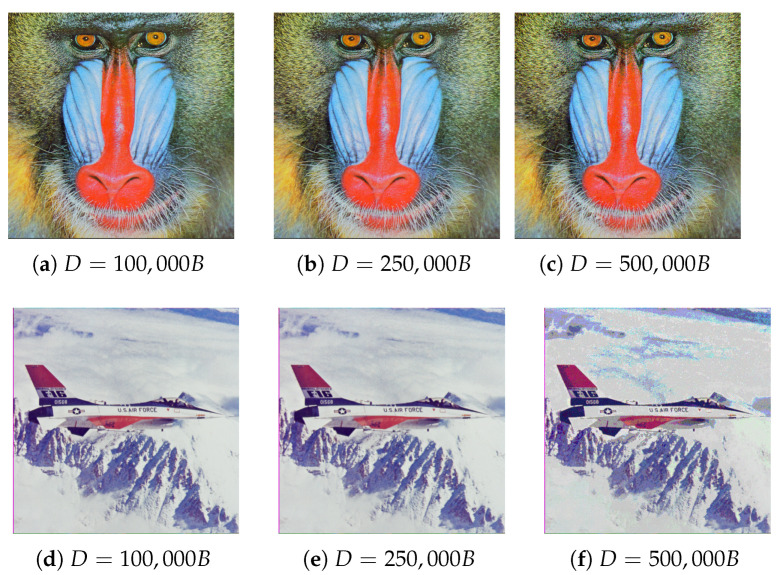
Steganogram comparison generated with given secret message capacity for vertex-based method (differences become visible when images are magnified significantly).

**Figure 11 entropy-25-00963-f011:**
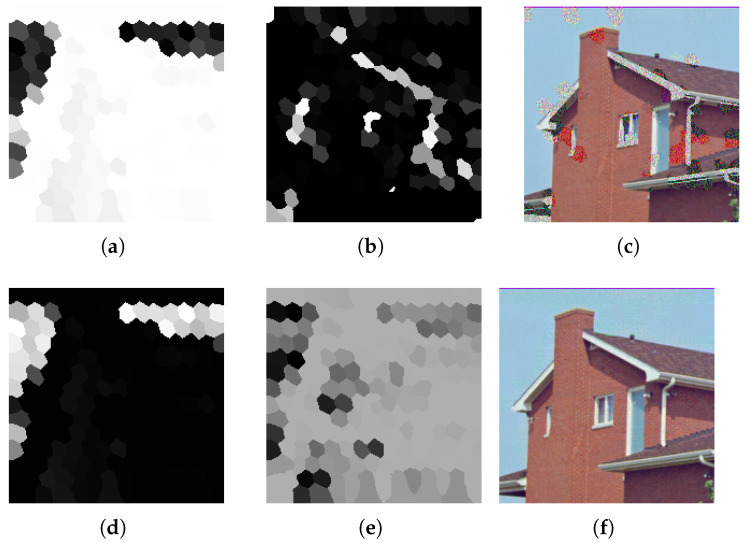
Comparison between image-to-graph conversion visualizations and masking matrices of obtained steganograms. The first row (**a**–**c**) contains images related to the process with distances inversely proportional to the pixel variance. The second row (**d**–**f**) presents the results if distances are proportional to variance. The first column contains image-to-graph conversion visualizations, the second presets generated masking matrices, and the last one contains the matrices after adjusting them to the same steganogram capacity of 50 kB.

**Figure 12 entropy-25-00963-f012:**
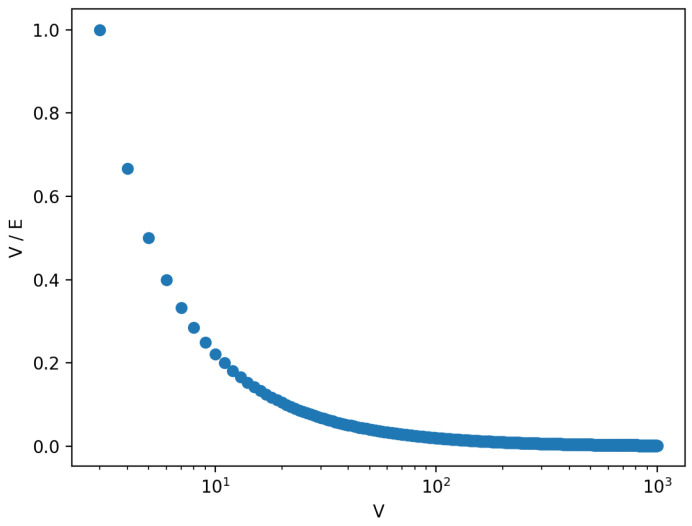
Chart expresses the ratio between the number of edges belonging to the Hamiltonian cycle of a fully connected graph of *V* vertices and *E* edges.

**Figure 13 entropy-25-00963-f013:**
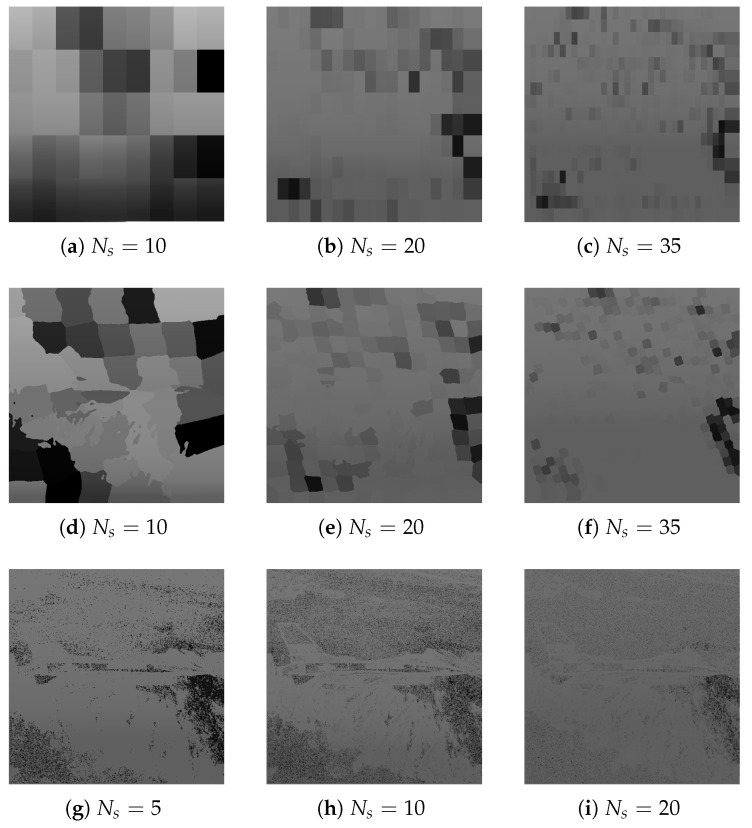
Masking matrices generated by the rectangle method (**a**–**c**), superpixels (**d**–**f**), and sets obtained using *k*-means (**g**–**i**).

**Figure 14 entropy-25-00963-f014:**
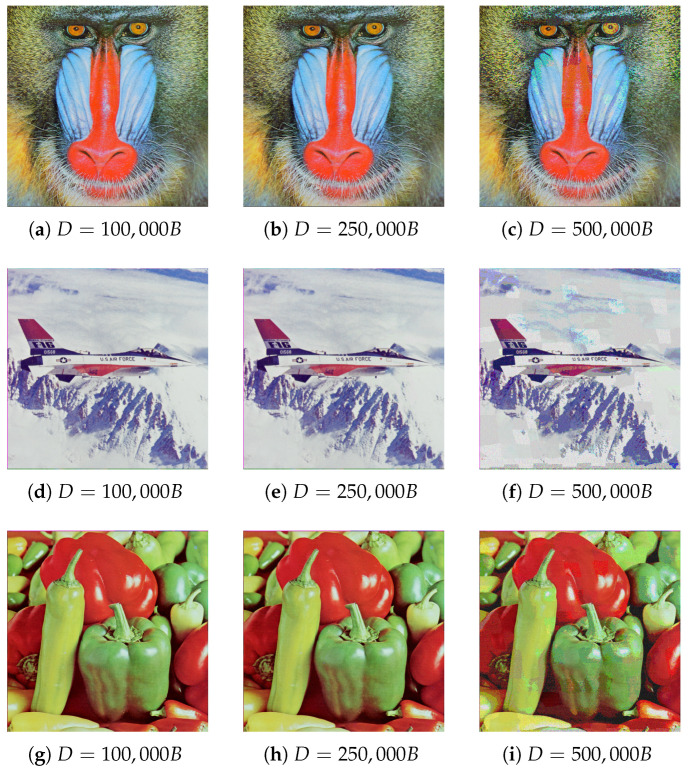
Comparison between steganograms with specified secret message capacity for edge-based method (differences become visible when images are magnified significantly).

**Table 1 entropy-25-00963-t001:** Steganogram quality measures depending on $s_{0}$. Bold values of specific metrics signify lowest distortion.

Scaling Factor $s_{0}$	*MSE*	*PSNR*	*SSIM*
0.25	6.93092	39.7568 dB	**0.952935**
0.50	6.67252	39.9218 dB	0.951723
0.75	6.62924	39.9501 dB	0.95122
1.00	**6.61124**	**39.9619 dB**	0.948736

**Table 2 entropy-25-00963-t002:** Results of the vertex based method.

Image	Target Capacity	*MSE*	*PSNR*	*SSIM*
512×512	[B]		[dB]	
Baboon	100,000 (12.71%)	0.53885	50.85	0.9988
Baboon	250,000 (31.78%)	6.60594	39.97	0.9900
Baboon	500,000 (63.57%)	272.732	23.81	0.7770
Airplane	100,000 (12.71%)	0.53689	50.87	0.9955
Airplane	250,000 (31.78%)	6.67252	39.92	0.9517
Airplane	500,000 (63.57%)	311.43	23.23	0.5165
Peppers	100,000 (12.71%)	0.53693	50.87	0.9960
Peppers	250,000 (31.78%)	6.71828	39.89	0.9582
Peppers	500,000 (63.57%)	292.99	23.50	0.5020

**Table 3 entropy-25-00963-t003:** Values of quality metrics depending on segmentation method (values in bold signify lowest quality degradation).

Segmentation Method	Number of Segments	*MSE*	*PSNR* [dB]	*SSIM*
Rectangular	10 (V=5)	1764.78	15.6978	0.186649
segmentation	21 (V=7)	1764.78	15.6978	0.186649
	45 (V=10)	27.2669	33.8084	0.891967
	190 (V=20)	7.25700	39.5572	**0.948116**
	595 (V=35)	6.85841	39.8025	0.947396
	1225 (V=50)	**6.85694**	**39.8034**	0.947764
Superpixels	10 (V=5)	1764.78	15.6978	0.186649
	21 (V=7)	21.7769	34.7848	0.903446
	45 (V=10)	19.2949	35.3103	0.913704
	190 (V=20)	7.07101	39.6699	**0.948129**
	595 (V=35)	6.90553	39.7728	0.946333
	1225 (V=50)	**6.77511**	**39.8556**	0.945946
*k*-means	10 (V=5)	7.89710	39.1901	0.940743
clustering	21 (V=7)	7.63390	39.3373	0.94536
	45 (V=10)	7.30898	39.5262	0.948244
	190 (V=20)	6.90061	39.7759	0.948481
	595 (V=35)	**6.76695**	**39.8608**	**0.949062**
	1225 (V=50)	-	-	-

**Table 4 entropy-25-00963-t004:** Results obtained using the edge based method.

Image	Target Capacity	*MSE*	*PSNR*	*SSIM*
512×512	[B]		[dB]	
Baboon	100,000 (12.71%)	0.89379	48.6524	0.9984
Baboon	250,000 (31.78%)	7.52079	39.4021	0.9877
Baboon	500,000 (63.57%)	663.829	19.9442	0.7236
Airplane	100,000 (12.71%)	0.58925	50.4617	0.9953
Airplane	250,000 (31.78%)	7.07101	39.6699	0.9481
Airplane	500,000 (63.57%)	362.857	22.5674	0.5105
Peppers	100,000 (12.71%)	0.58662	50.4812	0.9955
Peppers	250,000 (31.78%)	7.21685	39.5813	0.9520
Peppers	500,000 (63.57%)	411.621	22.0198	0.4787

**Table 5 entropy-25-00963-t005:** Result comparison with *LSB* method described in [37].

	Capacity			Least Significant Bit			Vertex Based Method			Edge Based Method		
Image	bpp	%	B	MSE	PSNR	SSIM	MSE	PSNR	SSIM	MSE	PSNR	SSIM
Airplane	2.67	11.12%	87,491 B	-	51.65 dB	**0.9970**	0.447	**51.66 dB**	0.9963	0.448	51.65 dB	0.9963
Baboon	2.67	11.12%	87,491 B	-	**51.65 dB**	**0.9997**	0.449	51.64 dB	0.9990	0.816	49.05 dB	0.9985
Peppers	2.67	11.12%	87,491 B	-	51.62 dB	**0.9998**	0.448	51.65 dB	0.9968	0.447	**51.66 dB**	0.9967

**Table 6 entropy-25-00963-t006:** Result comparison with *PVD* method described in [37].

	Capacity			Pixel Value Differencing			Vertex Based Method			Edge Based Method		
Image	bpp	%	B	MSE	PSNR	SSIM	MSE	PSNR	SSIM	MSE	PSNR	SSIM
Airplane	4.6699	19.45%	153,023 B	-	40.61 dB	0.9751	1.596	**46.13 dB**	**0.9879**	1.645	46.00 dB	0.9865
Baboon	5.3627	22.34%	175,725 B	-	37.83 dB	0.9945	2.012	**45.13 dB**	**0.9960**	2.647	43.94 dB	0.9953
Peppers	4.7177	19.65%	154,590 B	-	40.93 dB	**0.9980**	1.638	**46.21 dB**	0.9892	1.684	45.90 dB	0.9878

**Table 7 entropy-25-00963-t007:** Result comparison with *PSO-IWT* method described in [18].

	Capacity	*PSO-IWT*			Vertex Based Method			Edge Based Method		
Image	B	MSE	PSNR	SSIM	MSE	PSNR	SSIM	MSE	PSNR	SSIM
Airplane	73,728 B	-	41.13 dB	-	0.378	**52.39 dB**	0.9971	0.378	52.39 dB	0.9968
Baboon	73,728 B	-	41.38 dB	-	0.379	**52.38 dB**	0.9993	0.378	52.38 dB	0.9993

**Table 8 entropy-25-00963-t008:** Result comparison with *ACO* method described in [28].

	Capacity		*PSO-IWT*			Vertex Based Method			Edge Based Method		
Image	%	B	MSE	PSNR	SSIM	MSE	PSNR	SSIM	MSE	PSNR	SSIM
Baboon	2.8473%	22,396 B	0.8409	48.88 dB	-	0.1151	57.55 dB	0.9999	**0.1146**	**57.57 dB**	0.9998
Pepper	2.8661%	22,544 B	1.0605	47.88 dB	-	0.1151	**57.56 dB**	0.9994	**0.1150**	**57.56 dB**	0.9991

**Table 9 entropy-25-00963-t009:** Results of RS analysis.

Image	Method	Actual Capacity [B]	Estimated Capacity [B]	Relative Capacity Gain
Baboon	vertex	100,000	123,062	−18.74%
Baboon	vertex	250,000	157,644	+58.59%
Baboon	vertex	500,000	224,227	+122.99%
Airplane	vertex	100,000	85,480	+16.99%
Airplane	vertex	250,000	111,986	+123.24%
Airplane	vertex	500,000	205,120	+143.76%
Peppers	vertex	100,000	99,709	+0.29%
Peppers	vertex	250,000	108,834	+129.71%
Peppers	vertex	500,000	208,734	+139.54%
Baboon	edge	100,000	158,600	−36.95%
Baboon	edge	250,000	124,913	+100.14%
Baboon	edge	500,000	187,856	+166.16%
Airplane	edge	100,000	95,377	+4.85%
Airplane	edge	250,000	115,145	+117.12%
Airplane	edge	500,000	207,089	+141.44%
Peppers	edge	100,000	125,437	−20.28%
Peppers	edge	250,000	124,040	+101.55%
Peppers	edge	500,000	214,619	+132.97%

## Data Availability

Data is contained within this article.

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
