# Peer review of "Hiding Information in Digital Images Using Ant Algorithms"

_entropy, 2023, doi:10.3390/e25070963_

Round 1

Reviewer 1 Report

This article presents the idea of using the metaheuristic approach in stenography. The article's main focus is the problem of identifying the regions on the image, allowing the maximum amount of information to hide. The authors stated that the concept of using metaheuristics in stenography had been introduced previously. However, in this paper, the particular focus was put on region identification, which is a novelty in the field. The paper is well organized and presents sufficient background to follow the idea. However, I have some additional suggestions which could improve the overall readability of this work:

- first, some fragments (especially in section 2) could be omitted or at least shortened. For example, introducing a fragment in section 2 describing the process of concealing information in ancient Greece is unnecessary. The same applies to the last fragment in subsection 2.2.
- figures 1 and 2 are similar. I suggest leaving only a single figure, for example, related to data hiding, while only commenting on the similarity of the process in the case of data extraction;
- the title of algorithms 1 and 3 should be extended;
- a comma should end all equations;
- I would suggest moving subsection 5.2 before the experiments. Different metrics are used as part of the algorithm to evaluate the results.

The above suggestions are only minor remarks, and the paper's idea and the extended description of numerical experiments clearly show the advantages of the proposed methods. The authors also included comparisons with other methods found in the literature. Thus, the paper is well-presented and should be accepted. 

Reviewer 2 Report

In the reviewed paper authors proposed  an optimization of ant algorithms  in its application to information hiding problem for digital images.It was not the first attempt in the use ant algorithm for design of information hiding systems. It would be correctly to remember  references [37-39] in Abstract.

In the current paper are optimized the details of ant algorithm and its parameters in order to make a trade off between  embedding capacity (more common to say about embedding rate )  and image distortion.Unfortunately, this fact was not striked in Abstract,  but in introduction only) how it would be more appropriately to do. The main defect of the current paper is   (according the reviewer mind) the following : It is most commonly in steganography (SG) to use as the main criteria  of its quality ,   the notion of  security (confidentiality ) that can be presented, in turn,  by two probabilities : Pm     the probability   of stego system missing into the image and Pfa  --   the false alarm probability detection of SG presence  if, in fact, it was absent during detection procedure ! .Of course, image distortion should be also provided as minimal one in any case but it is not sufficiently for a providing of real SG security.This fact is remarked in the "main monograf" on SG (see J.Fridrich "Steganography in digital media " , Cambridge University Press, 2010).( By the way this book even was not cited in References of the reviewed paper...! ) There are a lot of scientific papers in which it is considered namely the notions of probabilities Pm and Pfa  as the main criteria of SG security (see, for example,S.Dumitrescu et al, "Detection of  LSB steganography via sample pair analysis", LNCS,2578,2002,p.355-372). It is obviously that    image distortion criterion if  image is slightly corrupted results in  SG detection .Therefore such criterion cannon be used  for professional SG. However , for particular cases when  in the role of SG detector can be assumed always a presence of some peoples, distortion criterion can be acceptable  . This is why  reviewer believes that the current paper still can be presented for publication in the journal "Entropy" but after elaboration taking into account  the following remarks:

1.Insert in Abstract a difference between the current paper and the previous published already papers where was proposed  application of ant algorithm to design of SG.

2.In introduction it is necessary to remark- what is the place of such image  distortion criterion like  confidentiality criterion of SG systems.

3 To short Introduction taking into account that the main notions concerning to SG are  well known for all researches which could be interested by direction of investigation in reviewed paper.

4.To  note in Introduction that cryptography can be also executed jointly with SG and by which means.

5.To indicate in the main text of the paper where was proved the formulas (1-3). 

6. Describe carefully in the main text ant algorithm itself. 

7 A comparing  of LSB method and method presented on the page description of experiments in the chapter 5  in the current paper is unclear that requires to specify this problem.

8. Description of experiments in the chapter 5 can be shorted because it is not necessary to present all  Fig 10,11 and 12 ; similar Tables  1-9 could be  presented partly,   removing results   that  can be obtained by similar manner.

9. It  would be useful to short the list of references  excluding multiple citing in text for some facts. 

10. It would be nice to renovate  list of References because about half of them have "age"  more than 10 years. 

Generally speaking, it seems may occur at most 20 pages in the full paper after such simplification.

No special comments except of some word absence  (see "of" on the page 14,"by" and "s" on the page 17 and check the correct use of some articles .

Reviewer 3 Report

The article is very well written overall. There are some clarifications that could be added to further improve the paper:

In subsection 4.2, the authors state that the masking matrix is dependent and very sensitive to changes of selected hyperparameters. Since the matrix helps to identify how much data should be inserted in various regions of the image, why is it tolerable to be very sensitive to changes in selected hyperparameter? Ideally, shouldn’t the masking matrix be unique for one image?

When presenting images like in Figure 12 or Figure 16, it is difficult to observe any meaningful information. The authors may consider replacing the representation with another, to better highlight differences. 

It is unclear what the subjective evaluation consisted of. The authors should better document this part of the evaluation, mentioning the number of subjects that participated and how the experiments were conducted. While a strong indicator of image distortion, PSNR is not a subjective measure.

The readability of the paper can be improved if the images are placed closer to where they are referenced in text. Also, splitting a paragraph with more than one page of images and tables should be avoided. The paper is very long, but the details are well presented, and the conducted activities can be reproduced.
